

# Algorithms for efficiently collapsing reads with Unique Molecular Identifiers

Daniel Liu

Torrey Pines High School, San Diego, CA, United States of America
Department of Psychiatry, University of California, San Diego, La Jolla, CA, United States of America

## ABSTRACT

**Background**. Unique Molecular Identifiers (UMI) are used in many experiments to find and remove PCR duplicates. There are many tools for solving the problem of deduplicating reads based on their finding reads with the same alignment coordinates and UMIs. However, many tools either cannot handle substitution errors, or require expensive pairwise UMI comparisons that do not efficiently scale to larger datasets.
**Results**. We reformulate the problem of deduplicating UMIs in a manner that enables optimizations to be made, and more efficient data structures to be used. We implement our data structures and optimizations in a tool called UMICollapse, which is able to deduplicate over one million unique UMIs of length 9 at a single alignment position in around 26 s, using only a single thread and much less than 10 GB of memory.
**Conclusions**. We present a new formulation of the UMI deduplication problem, and show that it can be solved faster, with more sophisticated data structures.

## INTRODUCTION

In many next-generation sequencing experiments, Unique Molecular Identifiers (UMI) are used to identify PCR duplicates (*Islam et al., 2014*; *Kivioja et al., 2012*). PCR amplification is typically a necessary step in sequencing as it increases the number of DNA fragments by duplicating each molecule multiple times. By ligating a short, random UMI sequence onto each strand of DNA fragment before PCR amplification, sequenced reads with the same UMI can be easily identified as PCR duplicates. This is because each random UMI is like a unique tag for each DNA molecule, which means that all duplicates of the same DNA molecule will contain the same UMI. UMIs are useful in many experiments, including RNA-seq (*Shiroguchi et al., 2012*), single-cell RNA-seq (scRNA-seq) (*Islam et al., 2014*; *Srivastava et al., 2019*), and individual-nucleotide resolution Cross-Linking and ImmunoPrecipitation (iCLIP) (*König et al., 2010*), as it allows PCR amplified reads that may cause inaccuracies in true DNA/RNA molecule counts to be collapsed before further downstream analysis. For example, in RNA-seq, UMIs can be used to accurately count and compare the number of RNA molecules for each sample (*Kivioja et al., 2012*). In scRNA-seq, it is possible to count the number of RNA molecules for each gene and cell combination by using UMIs (*Srivastava et al., 2019*). However, due to sequencing and PCR amplification errors, sequenced reads may contain UMI sequences that deviate from their

Corresponding author
Daniel Liu, daniel.liu02@gmail.com

true UMI sequences before PCR amplification (*Schirmer et al., 2015*). Therefore, software tools for accurately solving the deduplication task by grouping reads with the same true UMI must be able to handle errors in UMI sequences. We focus on tolerating substitution errors, and ignore the much rarer insertion and deletion errors, like previous works on the UMI deduplication task (*Smith, Heger & Sudbery, 2017*).

Before deduplicating sequenced reads by their UMI sequences, they must first be aligned to a reference genome or transcriptome, with the extracted UMI sequence stored in the read header. Then, the reads at each unique alignment location are independently deduplicated based on the UMI sequences. This is done by tools like UMI-tools (*Smith, Heger & Sudbery, 2017*), zUMIs (*Parekh et al., 2018*), umis (*umis, 2019*), gencore (*Chen et al., 2018*), fgbio (*fgbio, 2019*), Picard (*Picard Tools, 2019*), and Je (*Girardot et al., 2016*). It is also possible to skip the alignment step and deduplicate reads directly based on the entire DNA sequence, which includes the UMI (this is done by Calib (*Orabi et al., 2018*)). This does not require alignments to be explicitly computed, which may be faster on larger datasets. In either case, efficient methods for finding and grouping reads that are similar is needed. After grouping reads by their UMIs, we can report the reads in each of these groups directly, extract a consensus read for each group, or report the number of groups. Since these tasks are quite similar, we will only consider obtaining a consensus read for each group of reads while removing all other reads. This is usually done by just keeping the read that has the highest UMI frequency and the highest mapping quality or highest Phred quality scores in each group.

Though there have been many tools proposed for accurately handling UMI data, there have not been many works focused on discussing algorithms that make the UMI deduplication process more efficient in terms of speed and memory usage. Many previous tools either deduplicate purely based on unique UMIs without being error tolerant (*Mangul et al., 2017*; *Shiroguchi et al., 2012*), or require pairwise comparisons between UMIs (i.e., zUMIs (*Parekh et al., 2018*), Picard (*Picard Tools, 2019*), Je (*Girardot et al., 2016*), older versions of UMI-tools (*Smith, Heger & Sudbery, 2017*), gencore (*Chen et al., 2018*), and fgbio (*fgbio, 2019*)), which does not efficiently scale to larger datasets. *Smith, Heger & Sudbery (2017)* proposed network-based algorithms for estimating the actual number of UMI, which was implemented in their UMI-tools program. That allows it to avoid the issue of inaccurately overestimating the number of true UMIs from directly counting unique sequenced UMIs, a problem that arises from substitution errors in PCR amplification and sequencing.

We aim to make the UMI deduplication process more efficient, while still being error-tolerant against nucleotide substitutions, by examining algorithms and data structures that allow us to minimize the number of comparisons between UMIs, which is a necessary step in finding similar UMIs when substitution errors are allowed. For simplicity, we only consider the problem of deduplicating UMIs at a single alignment position. In other words, for an input list of UMIs from mapped reads at a single alignment coordinate, we wish to find a set of unique consensus UMIs, where each consensus UMI represents a group of reads that have the same UMI before sequencing and PCR duplication. By solving the UMI deduplication problem at a single alignment coordinate, we can trivially extend

it to multiple alignment coordinates or multiple cells (in scRNA-seq) by independently deduplicating the UMIs at each alignment coordinate and cell.

Our contributions in this paper are as follows:

- We generalize a time consuming subsection of the network-based algorithms used in UMI-tools (*Smith, Heger & Sudbery, 2017*) and other tools to a common interface, which allows different data structures to be substituted into the network-based algorithms. This allows us to formulate the problem in a way that enables optimizations to be made.
- We examine a wide variety of both previously proposed and novel data structures' theoretical and practical behaviors on large datasets, and show that we can achieve more efficient deduplication with the same general network-based deduplication algorithms as UMI-tools (*Smith, Heger & Sudbery, 2017*). These data structures are all implementations of our proposed interface.
- We implement our algorithms in Java, as both a library of algorithms and a command-line-based UMI collapsing tool called UMICollapse.

## METHODS

### Deduplication algorithms

Most deduplication algorithms attempt to build graphs, where the vertices are unique UMIs, and the edges that connect UMIs that are within a certain edit distance (*Smith, Heger & Sudbery, 2017*). Let $G = (V, E)$, which represents a graph that consists of $N$ total unique UMIs, where the length of each UMI is $M$. Also, let $f(v)$ represent the frequency of the UMI $v$, $\forall v \in V$. The frequency is obtained by examining the input list of all UMIs and counting the number of occurrences of each unique UMI. Each $u, v \in V$ is connected by an edge $(u, v) \in E$ iff $d(u, v) \le k$, where $d$ defines the Hamming distance function that allows substitution errors, and $k$ represents the maximum error/edit threshold. For two UMI sequences $a$ and $b$ of equal length $M$, the Hamming distance function is defined as

$$d(a, b) = \sum_{i=1}^{M} \delta(a_i, b_i)$$

where

$$\delta(x, y) = \begin{cases} 0 & \text{if } x = y \\ 1 & \text{if } x \ne y \end{cases}$$

and $a_i$ represents the $i$th character of the sequence $a$, $\forall i \in \{1 \dots M\}$.

There are three main deduplication algorithms for identifying groups of unique UMIs based on $G$: connected components, adjacency, and directional. These algorithms were discussed by *Smith, Heger & Sudbery (2017)* and implemented in UMI-tools. Other tools like zUMIs (*Parekh et al., 2018*) and fgbio (*fgbio, 2019*) also implement some or all of these algorithms. We show that all three methods can be formulated to make use of a standardized interface that is supported by some data structure that efficiently and implicitly represents the graph $G$. Such a data structure must implement the following two operations:

- REMOVE_NEAR $(u, k, F)$. Returns a set $S$ of all UMIs that are within $k$ edits from a given UMI $u$ $(d(u,v) \leq k)$, such that $f(v) \leq F$, $\forall v \in S$, $v \neq u$, and $v$ is not yet removed from the graph $G.u$ is known as the "queried UMI" in REMOVE_NEAR queries, and it is always included in $S$ if it is not removed. Also, set all UMIs in the set $S$ as removed. The frequency $F$ is necessary for the directional method of grouping UMIs, and UMIs are removed so they are not returned by future queries. The REMOVE_NEAR operation finds all vertices that are connected with $u$, without explicitly storing and building the graph $G$.

- CONTAINS($u$). Returns whether the UMI $u$ is not removed. Note that this operation can be trivially implemented for any data structure by keeping a separate hash table of UMIs that are not removed in $O(1)$ time, so we will not discuss this operation in detail.

---

**Algorithm 1** Finding connected components on a set of UMIs $V$.

---

**procedure** GET_CONNECTED_COMPONENTS($V, k$)
    Initialize a data structure that implements REMOVE_NEAR and CONTAINS with $V$ and $k$.
    $cc \leftarrow \{\}$                                ▷ Resulting set of connected components.
    **for** $u \in V$ **do**
        **if** CONTAINS($u$) **then**
            $cc \leftarrow cc \cup$ DFS_CC($u, k$)
        **end if**
    **end for**
    **return** $cc$
**end procedure**
**procedure** DFS_CC($u, k$)
    $c \leftarrow \{\}$                               ▷ Set of UMIs in a connected component.
    **for** $v \in$ REMOVE_NEAR($u, k, \infty$) **do**
        **if** $v \neq u$ **then**
            Add all elements of DFS_CC($v, k$) to $c$.
        **end if**
    **end for**
    **return** $c$
**end procedure**

---

This interface allows us to focus our efforts of optimizing REMOVE_NEAR queries with different data structures, while maintaining compatibility with the three algorithms for grouping UMIs. Instead of calculating an $N \times N$ matrix of edit distances between UMIs to build the graph $G$ like some previous tools (*Parekh et al., 2018*), we allow REMOVE_NEAR queries to dynamically change the underlying data structure that implicitly represents $G$ throughout the deduplication algorithms, which enables optimizations to be made. Note that for simplicity, we omit the necessary initialization step for any data structure that implements the interface. Next, we discuss the three algorithms for grouping UMIs in

detail, and show how these two operations are sufficient for implementing those three algorithms.

### Connected components

One method for identifying groups of unique UMIs is based on *connected components* (known as *clusters* in UMI-tools (*Smith, Heger & Sudbery, 2017*) and Calib (*Orabi et al., 2018*)). A connected component in a graph $G = (V, E)$ is a set of vertices $S \subseteq V$ where there is a valid path between each pair of vertices $u, v \in S$, and there is no edge $(u, v) \in E$ where $u \in S$, but $v \notin S$. Each connected component is a group of UMIs, and the UMI that has the highest frequency is chosen to represent the group. This algorithm allows UMIs that are similar under the Hamming distance metric to be grouped together, but it can lead to an underestimation of the total number UMIs. There may exist an UMI $w$ for two UMIs $u$ and $v$, where $d(u, w) \leq k$ and $d(w, v) \leq k$, causing $u$ and $v$ to be grouped together even though $d(u, v) > k$ (essentially "bridging" the Hamming distance "gap" between the two different UMIs). We show the entire connected components algorithm in Algorithm 1.

The time complexity of finding all connected components is $O(N)$ with respect to each query to REMOVE_NEAR, since only one REMOVE_NEAR query is allowed per vertex (UMI) in the graph, and each vertex is only visited once.

---

**Algorithm 2** The adjacency algorithm on a set of UMIs $V$.

---

**procedure** GET_GROUPS_ADJACENCY($V, k$)

    Initialize a data structure that implements REMOVE_NEAR and CONTAINS with $V$ and $k$.

    $adj \leftarrow \{\}$                         ▷ Resulting set of groups of adjacent UMIs.

    Sort $V$ by decreasing UMI frequency values.

    **for** $u \in V$ **do**

        **if** CONTAINS($u$) **then**

            $adj \leftarrow adj \cup$ REMOVE_NEAR($u, k, \infty$)

        **end if**

    **end for**

    **return** $adj$

**end procedure**

---

### Adjacency

To avoid the underestimation issue of the connected components algorithm, we can sort the UMIs by their frequencies, and examine UMIs from higher to lower frequency. We assume that UMIs that appear at a higher frequency are more likely to be correct, and allow adjacent UMIs in $G$ that have lower frequencies to be grouped with a higher frequency UMIs. Two UMIs $u$ and $v$ are adjacent if there exists an edge $(u, v) \in E$. The lower frequency UMIs are assumed to be due to substitution errors. The full algorithm is presented in Algorithm 2.

Since we must first sort the $N$ UMIs in $O(N \log N)$ time and then query REMOVE_NEAR at most $N$ times, the overall time complexity is $O(N \log N + NQ)$, where $Q$ is the time

complexity of REMOVE_NEAR. If $Q \geq \log N$, then the time complexity with respect to each query is $O(N)$.

### Directional

The directional algorithm allows UMIs that are not adjacent to be grouped together, and it also minimizes underestimation from examining connected components. Like the adjacency method, it involves iterating through the sorted UMIs from highest to lowest frequency. Instead of grouping together all adjacent vertices like in the adjacency method, we add a vertex $v$ that is adjacent to vertex $u$ to the group iff $f(v) \leq \epsilon[f(u) + 1]$, where $0 \leq \epsilon \leq 1$ and $\epsilon$ is a parameter that represents the threshold percentage at which adjacent UMIs are grouped, which is usually $\epsilon = 0.5$ (this is a rearrangement of UMI-tools' inequality $2f(v) - 1 \leq f(u)$ (*Smith, Heger & Sudbery, 2017*)). Then, the process is recursively repeated starting at $v$ by examining the adjacent vertices of $v$. This allows UMIs that are not directly adjacent to a high frequency UMI to be visited and grouped with the high frequency UMI if they have a low frequency. Algrorithm 3 represents the full algorithm.

---

**Algorithm 3** The directional algorithm on a set of UMIs $V$.

**procedure** GET_GROUPS_DIRECTIONAL($V, k, \epsilon$)

    Initialize a data structure that implements REMOVE_NEAR and CONTAINS with $V$ and $k$.

    $dir \leftarrow \{\}$         ▷ Resulting set of groups of UMIs.

    Sort $V$ by decreasing UMI frequency values.

    **for** $u \in V$ **do**

        **if** CONTAINS($u$) **then**

            $dir \leftarrow dir \cup$ DFS_DIR($u, k, \epsilon$)

        **end if**

    **end for**

    **return** $dir$

**end procedure**

**procedure** DFS_DIR($u, k, \epsilon$)

    $g \leftarrow \{\}$         ▷ Group of UMIs found by the directional algorithm.

    **for** $v \in$ REMOVE_NEAR($u, k, \epsilon[f(u) - 1]$) **do**

        **if** $v \neq u$ **then**

            Add all elements of DFS_DIR($v, k, \epsilon$) to $g$.

        **end if**

    **end for**

    **return** $g$

**end procedure**

---

The overall time complexity of directional is $O(N)$, which is similar to that of adjacency, since they both require an initial sorting of the $N$ UMIs, and each UMI requires a REMOVE_NEAR query.
## Efficient methods for deduplication

A naive implementation of REMOVE_NEAR runs in $O(NM)$ time in the worst case, by examining each of the $N$ UMIs and calculating the Hamming distance in $O(M)$ time for each UMI. As we have shown that each of the network-based grouping algorithms require $O(N)$ time with respect to each REMOVE_NEAR query, the overall run time complexity is $O(N^2 M)$. This scales quadratically with the total number of UMIs, and it becomes prohibitively slow as $N$ increases, making it inadequate for handling larger sets of data. However, there are four properties of the REMOVE_NEAR function and the UMI sequences in general that allow for optimization:

- We do not care about the exact number of edits between two UMI sequences, only that their Hamming distance is less than the threshold $k$.
- We do not need to reexamine previously removed UMIs when there are multiple, successive REMOVE_NEAR queries. This benefit is due to how we frame the deduplication problem in a way that makes use of REMOVE_NEAR queries that dynamically queries/updates the implicit representation of the UMI graph $G$ instead of outright constructing the entire graph $G$.
- We do not need to examine UMIs that appear at a higher frequency than the threshold $F$.
- UMIs sequences are generally very short. It is not unreasonable to assume that they are $\leq 20$ nucleotides long in most cases.

We examine many different methods for efficiently handling queries to the REMOVE_NEAR function by making use of these four properties. Most of them are special data structures that speed up REMOVE_NEAR queries. Note that most data structures depend on the number of resulting UMIs from the REMOVE_NEAR query, which may vary depending on the degrees of vertices in the graph $G$.

### Constant-time Hamming distance calculations

Let $\oplus$ denote the bitwise XOR operation, and let the POP_COUNT function return the number of set (1) bits in a binary string. For two binary strings $a$ and $b$ where $a, b \in \{0, 1\}^M$, the Hamming distance (total number of mismatches) between them can be easily calculated by $pop\_count(a \oplus b)$. This operation can be done in $O(1)$ time if both $a$ and $b$ are shorter than the word length in a computer, which is typically 64-bits. We show that this idea can be extended to all four nucleotides ({A, T, C, G}).

The main property we want for an encoding of the four nucleotides is that the encodings are pairwise equidistant. One such set of encodings with the shortest possible length for each encoding is

$$e_A = 110 \quad e_T = 011$$
$$e_C = 101 \quad e_G = 000$$

These encodings can be viewed as the vertices in a regular 3-simplex (tetrahedron) in Hamming space, and they are discussed in *Liu (2019)*. The Hamming distance between each pair of different encodings is exactly two, which means that for two UMI sequences

$u$ and $v$, we can translate them into bit strings $a$ and $b$ using the encoding method above, and the overall Hamming distance between the two bit strings is exactly $2d(a,b)$. This can be used to easily infer $d(a,b)$. For example, let "AAT" and "AAA" map to 110110011 (concatenate $e_A e_A e_T$) and 110110110 (concatenate $e_A e_A e_A$), respectively. Then, $pop\_count(110110011 \oplus 110110110) = 2d(\text{AAT}, \text{AAA}) = 2$. Thus, the edit distance $d(\text{AAT}, \text{AAA}) = 1$.

Encoding each UMI sequence saves time and memory, since multiple encoded nucleotides can be stored in a computer word, and the Hamming distance between two UMIs can be calculated in constant time per word by using the XOR and POP_COUNT operations. $\lfloor \frac{64}{3} \rfloor = 21$ nucleotides can be packed into one computer word of length 64, and an array of words can be used to handle longer UMI lengths. Note that we separately encode the positions of undetermined N nucleotides in sequences that contain it. We treat this undetermined nucleotide as different from the other four nucleotides.

For all other data structures and algorithms that we discuss, we assume that the UMI lengths are short, and Hamming distances are calculated with this method, which takes $O(1)$ time per calculation. This also speeds up hash operations and equality comparisons between UMI sequences to $O(1)$ time, since they can process the entire encoded UMI sequence at once.

### Combinations

Instead of a brute-force search through all $N$ UMIs during REMOVE_NEAR queries, it may be faster to directly generate UMIs sequences that are within a certain edit distance and checking whether each of them exists using a hash table. The run time complexity of this "combinations" algorithm, for each REMOVE_NEAR query with an alphabet $\Sigma = \{A, T, C, G\}$ of length $|\Sigma|$, is $O(|\Sigma|^k \binom{M}{k}) = O(|\Sigma|^k M^k)$. This is because any $k$ nucleotides in the UMI can be edited, and each nucleotide can be edited to any nucleotide in $\Sigma$. Each combination can be checked in $O(1)$ time on average with the hash table that contains all $N$ UMIs. This method is efficient if $k$ is small, but it scales exponentially with $k$, and it does not make use of our constant-time Hamming distance calculations. This method is used in *umis (2019)* for UMI deduplication.

### Trie

The main problem with the combinations algorithm is that it may examine multiple UMI combinations that do not actually exist in the input UMIs sequences. To address this problem, a string trie (*De La Briandais, 1959*) can be built to store each of the $N$ UMIs. Each node in a trie consists of up to $|\Sigma|$ children—one for each character in $\Sigma$. Also, each unique path from the root of the trie to a leaf node at depth $M$ represents a unique UMI of length $M$, with each node on the path representing a nucleotide in the UMI sequence. A trie allows prefixes that are shared among multiple sequences to be collapsed into one prefix path, which saves space. However, the main benefit of a trie is that while generating UMI combinations, the current prefix of the UMI combination can be checked against nodes that exist in the trie. This allows prefixes of UMI combination that cannot possibly construct an existing UMI to be pruned, which speeds up queries in practice. An example of a trie is shown in Fig. 1.

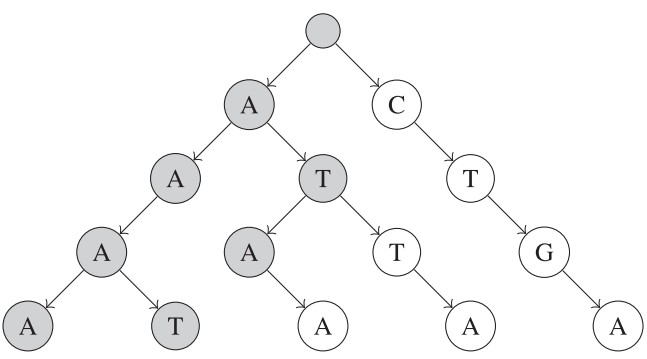

**Figure 1  Simplified example of a trie built on five UMIs.** These UMIs are "AAAA", "ATAA", "CTGA", "AAAT", and "ATTA". Only nodes that are shaded are visited when searching for UMIs within 1 edit of "AAAG".

To build the trie, around $MN$ operations are necessary to insert every UMI sequence into the trie, for a total of $O(MN)$ time. Each REMOVE_NEAR query will take at least $O(MR)$ time, if $R$ total UMI sequences are returned by the query, and the exact time taken depends on how many UMIs share prefixes and $k$, the number of edits allowed. The factor of $M$ is due to the trie's height being exactly $M$ nodes.

Some subtrees in the trie may become completely removed after some REMOVE_NEAR queries, when all UMIs in that subtree are removed. These subtrees can be completely skipped during future REMOVE_NEAR queries to save time while processing future queries, by keeping track of a flag that indicates whether the subtree of each node still exists. This flag can be recursively propagated up towards the root of the trie during REMOVE_NEAR queries (i.e., each node's subtree is completely removed iff all children subtrees of that node are completely removed). Then, an entire subtree can be pruned if the "completely removed" flag is set, during REMOVE_NEAR queries.

Similarly, we can keep track of the minimum frequency of any UMI within each subtree of the trie. After each query and removal, the minimum frequency of each node's subtree can be recursively updated for non-leaf nodes, to exclude the UMI frequency of any removed UMIs (i.e., each node's subtree's minimum frequency is the minimum across the minimum frequencies of that node's children subtrees that are not completely removed). This allows an entire subtree of the trie to be discounted if its minimum frequency is higher than $F$, since that implies that no UMI within the subtree has a lower UMI frequency than the maximum threshold $F$.

### n-grams

An alternative method for speeding up the naive $O(N)$ time REMOVE_NEAR query is by filtering UMIs based on partial exact matches of UMI sequences. The benefit of exact matches is that they can be efficiently indexed in a hash table, and therefore search operations would take $O(1)$ time on average. The general idea of the $n$-gram algorithm is to first decompose each UMI sequence into contiguous, non-overlapping sequences ($n$-grams) of length $n$ during the initialization of the $n$-gram hash table. Each unique

*n*-gram, which is represented by both its location in an UMI sequence and its actual nucleotide sequence, maps to a bin (list) of UMI sequences that contains the *n*-gram at its specific location in the UMI sequence. Then, to search for all UMIs within $k$ Hamming distance away from the queried UMI sequence, we can first decompose the query sequence into *n*-grams with the same method used during initialization, and only search the bins that corresponded to the *n*-grams created from the queried UMI. This is used by later versions of UMI-tools (*Smith, Heger & Sudbery, 2017*) to speed up the construction of its networks for UMI deduplication.

The UMI sequences must be split into $k+1$ contiguous and non-overlapping *n*-grams, where the *n*-grams are all approximately equal in length. For two UMI sequences that are within $k$ edits apart, and are split into $k+1$ *n*-grams, there must always be at least one *n*-gram that exactly matches in both sequences. This must be true, since in the "worst-case scenario" when there are exactly $k$ edits and $k$ of the *n*-grams have an edit within them somewhere, there is still exactly one *n*-gram that perfectly matches since it must have no edits due to the edit threshold of $k$. We also want to maximize the length of each individual *n*-gram, in order to maximize the number of distinct *n*-grams and prune more of the search space. Therefore, the length of each *n*-gram must be $\lfloor \frac{M}{k+1} \rfloor$, except for the last *n*-gram, which may be slightly longer if $M \bmod (k+1) \neq 0$. Implementation-wise, each *n*-gram is represented as a *view* on a portion of a UMI sequence, to avoid creating unnecessary copies of the UMI sequence data.

When initializing the hash table of UMI *n*-grams, the time complexity for $N$ UMIs is simply $O(MN)$, since each *n*-gram of each UMI sequence must be extracted and hashed. To estimate the expected run time of each REMOVE_NEAR query, we assume that the UMIs are uniformly sampled across the sequence space $\Sigma^M$. Then, for each possible *n*-gram segment location, there are around

$$N \frac{|\Sigma|^{M - \frac{M}{k+1}}}{|\Sigma|^M} = \frac{N}{|\Sigma|^{\frac{M}{k+1}}}$$

different UMI sequences at each *n*-gram location, assuming $M$ is divisible by $k+1$. Therefore, the time complexity of each query is $O(M + kN|\Sigma|^{-M/k})$ on average, since each bin of the $k+1$ *n*-grams of the queried UMI must be examined. This algorithm scales very well with longer UMI lengths, since the number of possible *n*-gram combinations increases, which decreases the likelihood of two UMI sequences having the same *n*-gram and allows each *n*-gram bin to be smaller.

### Subsequences

Another method for decomposing a UMI sequence into a more general representation during both initialization and queries is by extract subsequences of length $M - k$ (*SymSpell, 2019*). Alternatively, this can be thought of as picking any $k$ nucleotides in the UMI, and replacing all of them with a new placeholder character (i.e., a subsequence of "ATCG" is "A★CG" if $k = 1$ and the placeholder is ★). It is easy to see that if two subsequences with exactly $k$ placeholder characters exactly match, then their corresponding UMIs must match within $k$ edits. After a hash table that is indexed by these subsequences is built in

$O(N\binom{M}{k}) = O(M^k N)$ time, we only need to examine the $\binom{M}{k}$ bins of UMIs that correspond to each of the subsequences of the queried UMI during a query, in $O(M^k + R)$ time, to get $R$ UMI results. Note that some UMIs may have to be examined multiple times during a single query, since each UMI is placed in $\binom{M}{k}$ different bins when building the hash table. This is faster than the combinations algorithm because it spreads out the work of generating sequence combinations within $k$ edits over the initialization and the queries phases of the algorithm. This method is implemented in Calib (*Orabi et al., 2018*) and referred to as "locality-sensitive hashing".

---

**Algorithm 4** The REMOVE_NEAR operation for the BK-tree data structure.

---

**procedure** REMOVE_NEAR($u, k, F$)
    **return** DFS_BK_TREE($root, u, k, F$)          ▷ Start at the root node of the BK-tree.
**end procedure**
**procedure** DFS_BK_TREE($c, u, k, F$)
    $r \leftarrow \{\}$                          ▷ Resulting list of removed UMIs.
    $\Delta \leftarrow d(c, u)$
    **if** $c$ is not removed and $\Delta \leq k$ and $f(c) \leq F$ **then**
        Remove $c$.
        $r \leftarrow r \cup c$
    **end if**
    **for** each child $v$ of node $c$ where $\Delta - k \leq d(c, v) \leq \Delta + k$ **do**          ▷ Pruning by Hamming distance.
        **if** $\exists w$ in the subtree of $v$, where $w$ is not removed **then**     ▷ Pruning by removed nodes
            **if** $\exists w$ in the subtree of $v$, where $f(w) \leq F$ **then**       ▷ Pruning by frequency threshold $F$
                Add each element in DFS_BK_TREE($v, u, k, F$) to $r$.
            **end if**
        **end if**
    **end for**
    **return** $r$
**end procedure**

---

### BK-Tree

The BK-tree (*Burkhard & Keller, 1973*) is a type of metric tree that can make use of constant-time Hamming distance computations for handling REMOVE_NEAR queries. Each node in a BK-tree represents an UMI sequence, and each child node in a BK-tree is indexed by the Hamming distance between that child node's UMI and its parent node's UMI in an array in the parent node. When inserting an UMI, if a parent node $v$ already has a child node at a specific Hamming distance $d(u, v)$ for some inserted UMI $u$, then the insertion operation is recursively continued with the child node as the new parent node. Otherwise,
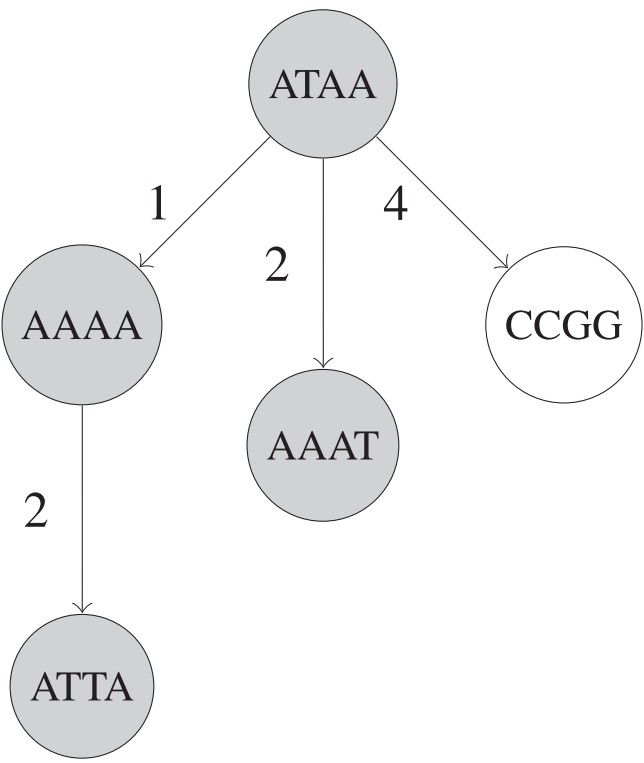

**Figure 2  Simplified example of a BK-tree built on five UMIs.** These UMIs are "ATAA", "AAAA", "CCGG", "AAAT", and "ATTA", inserted in that order into the tree. Only nodes that are shaded are visited when searching for UMIs within 1 edit of "AAAG". Each edge between two UMIs $u$ and $v$ has a number that indicates $d(u,v)$, the Hamming distance between $u$ and $v$.

a new node with the $u$ is created and attached to the parent node at the index $d(u,v)$ in parent node's array. A fully built BK-tree is shown in Fig. 2.

When querying, we recursively visit a subset of nodes in the BK-tree. For each node we visit starting from the root, we find the edit distance $d(u,v)$ between the queried UMI $u$ and the UMI at current node $v$. Then, we only visit the $2k+1$ children of the node of $v$ that are indexed by some edit distance $i$, where $d(u,v)-k \leq i \leq d(u,v)+k$. This is due to the triangle inequality that mandates that

$$d(u,w)+d(v,w) \geq d(u,v),$$
$$d(u,v)+d(u,w) \geq d(v,w)$$

for some node $w$ in the subtree of $v$ which is considered as a candidate for the query's result. Note that $i = d(v,w)$. Rearranging, we get

$$d(u,w) \geq d(u,v)-d(v,w),$$
$$d(u,w) \geq d(v,w)-d(u,v)$$

Since we only care about a candidate node $w$ if $k \geq d(u,w)$, we can substitute and arrive at

$$k \geq d(u,v)-d(v,w),$$
$$k \geq d(v,w)-d(u,v)$$

which can be rearranged to obtain

$$i = d(v, w) \geq d(u, v) - k,$$
$$i = d(v, w) \leq d(u, v) + k$$

This allows us to restrict the number of nodes visited during a query operation and speed up the query (*Burkhard & Keller, 1973*).

The time complexity of initializing and querying a BK-tree depends heavily on the height of the tree. On average, inserting all UMIs during initialization requires $O(N \log N)$ time, since the height of the tree is around $O(\log N)$. For each query that results in $R$ UMIs in total, the average time complexity is $O(kR + k \log N)$. The factor of $k$ is due to extra children nodes that are examined while traversing the tree. BK-trees are efficient since the height of a BK-tree built on random UMIs is expected to be very low compared to the total number of unique UMI sequences.

If all nodes in a subtree are removed, then that entire subtree can be skipped during each REMOVE_NEAR query, which saves some time. To keep track of these subtrees, we can propagate a flag that indicates whether an entire subtree is completely removed, from child nodes to parent nodes (i.e., a node's subtree is completely removed iff that node is removed and all of its children subtrees are completely removed).

Similarly, if all nodes in a subtree have a UMI frequency greater than the threshold $F$, then that subtree does not need to be examined in the query. The minimum frequency of each subtree can be stored and updated after each REMOVE_NEAR operation to prune subtrees from the search space (i.e., a node's subtree's minimum frequency is the minimum of that node's UMI frequency if it is not removed, and the minimum frequencies of each of the node's children subtrees that are not completely removed). Subtrees that have a minimum frequency greater than $F$ are pruned since they have cannot contain any node that has a UMI frequency less than or equal to the threshold $F$.

Typically, nodes are inserted in arbitrary order during initialization. However, to ensure that more subtrees are pruned by their minimum frequencies, the UMIs can be inserted in increasing order, based on the frequency of the UMIs. By inserting higher frequency UMIs later, they end up closer to the leaf nodes of the tree, and allow lower frequency UMIs to end up closer to the root of the tree. This allows more subtrees to be pruned by the frequency threshold $F$.

A sketch of the full algorithm for handling REMOVE_NEAR queries with a BK-tree is presented in Algorithm 4.

### Fenwick BK-trees

In REMOVE_NEAR queries, if the number of high frequency UMIs significantly outnumber the number of low frequency UMIs, it may be beneficial to first search for all UMIs that have a frequency lower than $F$, and then examine those UMIs to find ones that are within $k$ edits of the queried UMI. This is an alternative approach for solving REMOVE_NEAR queries compared to other methods like directly using BK-trees (*Burkhard & Keller, 1973*), since we build an overarching data structure for pruning UMIs based on UMI frequencies instead of edit distance. Finding UMIs with frequencies lower than a threshold can be done with a

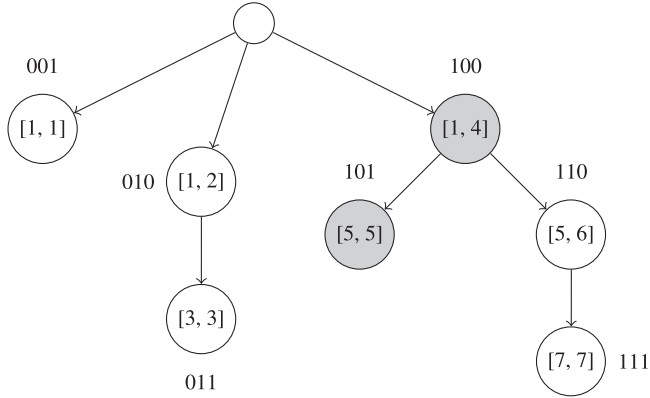

**Figure 3  Example of a Fenwick tree on UMI frequencies in the interval [1, 7].** Each node contains a BK-tree that has all UMIs sequences with frequency values in the interval that node describes. The binary labels for each node are used to implicitly index the Fenwick tree's nodes in a flat 1D array. As an example query, shaded nodes in the Fenwick tree must be visited to find UMIs that have frequencies $\leq 5$ and are within a certain edit distance from the queried UMI. Each of those nodes' BK-trees must be queried to find UMI sequences.

Fenwick tree (*Fenwick, 1994*) on an array of BK-trees that is indexed by UMI frequencies. A Fenwick tree allows for fast queries on a prefix of an underlying array described by the Fenwick tree, as each node in the Fenwick tree describes a range of elements in the array, and only $\log N$ nodes are visited per prefix query. Figure 3 visualizes a Fenwick tree during a query operation. If the underlying array is indexed by the frequencies of the UMIs, then a prefix query up to index $F$ on the Fenwick tree will return some property for all frequencies $\leq F$. In our case, each element in the underlying array is a BK-tree, and thus each node in the Fenwick tree also contains a BK-tree. The BK-tree of a parent node in the Fenwick tree will include UMI sequences from the BK-trees of each of the child nodes. When querying, BK-trees in the nodes of the Fenwick tree that represent the prefix up to $F$ in the frequency array are independently traversed to find UMIs that are within $k$ edits of the queried UMI.

To insert an UMI sequence, we must update $\log N$ BK-trees, each of height $\log N$ on average. Thus, initializing the Fenwick BK-trees data structure requires $O(N\log^2 N)$ time. Each REMOVE_NEAR query will take $O(kR + k\log^2 N)$ time, since $\log N$ BK-trees must be examined. However, in practice, we expect the heights of the BK-trees to be very short, and we expect this method to prune a significant portion of the UMIs based on their frequencies.

### n-grams BK-trees

We propose a new method for faster REMOVE_NEAR queries based on a combination of BK-trees (*Burkhard & Keller, 1973*) and *n*-grams. The original *n*-gram algorithm requires a linear scan through each UMI that shares an *n*-gram with the queried UMI. We replace the bin of corresponding UMIs for each *n*-gram with separate BK-tree to speed up queries. This allows the height of each BK-tree to be very short, and also allows us to make use of methods for pruning subtrees, like ignoring subtrees that are completely removed and subtrees with minimum frequencies larger than $F$. Overall, this method is faster than using

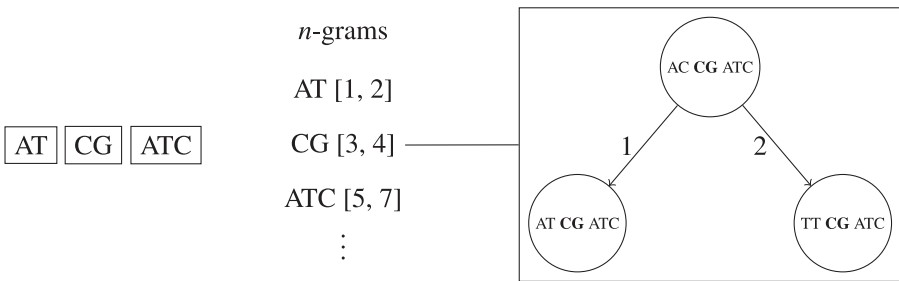

**Figure 4 Simplified example of the *n*-grams BK-trees data structure.** If two edits are allowed, then each UMI is split into three non-overlapping *n*-grams. Each *n*-gram sequence and its interval position in the full UMI sequence corresponds to a BK-tree that contains all UMIs that the same *n*-gram sequence at the same position.

just the *n*-grams method on average, taking only

$$
O\left( M + kR_1 + k\log\left(\frac{N}{|\Sigma|^{\frac{M}{k}}}\right)\right.
$$

$$
\left. + kR_2 + k\log\left(\frac{N}{|\Sigma|^{\frac{M}{k}}}\right) + \ldots + kR_{k+1} + k\log\left(\frac{N}{|\Sigma|^{\frac{M}{k}}}\right)\right)
$$

$$
= O\left( M + kR + k^2\log\left(\frac{N}{|\Sigma|^{\frac{M}{k}}}\right)\right)
$$

time, where the sum of the number of query results across each unique *n*-gram's corresponding BK-tree is $R$ (i.e., $R_1 + R_2 + \ldots + R_{k+1} = R$), and each $O(\log(N|\Sigma|^{-M/k}))$ is due to the height of the BK-tree corresponding to one of the $k+1$ unique *n*-gram of length $\lfloor\frac{M}{k+1}\rfloor$. If a significant portion of the UMI sequences share the same *n*-gram, then the algorithms will run as fast as just using one large BK-tree for all UMI sequences, which takes $O(kR + k\log N)$ time, not $O(N)$ time. This method is visualized in Fig. 4.

## Implementation

We implement all of the algorithms and data structures discussed in a proof-of-concept Java program called UMICollapse. In our implementation, the main focus was efficiency, so the breadth of features in UMICollapse is narrower than that of other popular tools (*Smith, Heger & Sudbery, 2017*; *Srivastava et al., 2019*). Since each data structure and deduplication algorithm is modular and self-contained, UMICollapse can be easily used as a library and imported into other projects. UMICollapse provides two main features: deduplicating raw FASTQ reads based solely on the read sequences, and deduplicating aligned SAM/BAM files. In a SAM/BAM file, reads at the same alignment coordinate (for forwards reads, the alignment start, and for reversed reads, the alignment end) are deduplicated based on their UMIs, which are stored in the read headers during preprocessing prior to alignment. This is implemented similarly to how UMI-tools (*Smith, Heger & Sudbery, 2017*) works, in order to simplify integration into existing UMI pipelines. UMICollapse chooses consensus reads from each group of reads with similar UMIs based on UMI frequency and alignment
**Table 1 Time taken for different algorithms to run as the number of unique UMIs ($N$) changes.** Each run time is measured in milliseconds and averaged over three trials. The best time for each dataset is bolded. Empty cells indicate algorithms that took longer than 10 min total to run one warm-up trial and three actual trials. "Combo" represents the combinations method, "Subseq." represents the subsequences method, and "BK-tree" is shortened to just "BK".

| Unique UMIs | Naive | Combo | Subseq. | Trie | $n$-grams | BK-tree | Fenwick BK | $n$-grams BK |
|---|---|---|---|---|---|---|---|---|
| 1,675 | 43 | 123 | 33 | 29 | 12 | 15 | 22 | **8** |
| 16,878 | 5,403 | 366 | 151 | 191 | 76 | 353 | 769 | **66** |
| 167,578 | – | 3,840 | 2,252 | 2,680 | 2,127 | 21,623 | 50,670 | **1,102** |
| 1,550,871 | – | 42,155 | 31,572 | 34,979 | – | – | – | **28,069** |

quality or Phred quality scores. Currently, UMICollapse can only handle single-end, reads and it cannot separately collapse reads per cell or per gene, which is useful for deduplicating scRNA-seq data.

The Java source code of UMICollapse and other simulation data generation programs is available at https://github.com/Daniel-Liu-c0deb0t/UMICollapse under the MIT License.

## RESULTS

All of the experiment were done on a laptop computer with a 2.7 GHz Intel Core i7-7500U CPU. The Java Virtual Machine (JVM) was limited to 8 GB of RAM when running UMICollapse, and UMI-tools (*Smith, Heger & Sudbery, 2017*) (version 1.0.0) was limited to around 10 GB of RAM. Some experiments had a maximum time threshold, where they were terminated if they took too long. Default settings were used where possible when running both tools.

### Comparing data structures

We compare each data structure's performance on handling REMOVE_NEAR queries as the number of unique UMIs ($N$), the length of UMIs ($M$), and the Hamming distance threshold ($k$) changes. To do so, we simulate UMI datasets represented by a tuple ($C$, $M$, $k$), where $C$ represents the number of "center" UMIs and we use the nucleotides $\Sigma = \{A, T, C, G, N\}$. Simulating a ($C$, $M$, $k$) dataset involves first generating $C$ random center UMIs of length $M$. Then, for each center UMI, we generate 20 random UMIs of length $M$ that are within $k$ edits of the center UMI. Each center UMI is assigned a "higher" random frequency, and each of the other UMIs is assigned a "lower" random frequency. In other words, any pair of center UMI $u$ and non-center UMI $v$ must satisfy the following inequality: $2f(v) - 1 \leq f(u)$. The set of unique UMIs and their corresponding frequencies simulated using this process represents a ($C$, $M$, $k$) dataset. Our tests are all based on the task of deduplicating a dataset of unique UMIs using the directional algorithm.

We generate four datasets to test the performance of different data structures as the number of unique UMIs increases. These four ($10^2$, 10, 1), ($10^3$, 10, 1), ($10^4$, 10, 1), and ($10^5$, 10, 1) datasets each have 1,675, 16,878, 167,578, and 1,550,871 unique UMIs, respectively. The run time of different data structures on these four datasets is shown in Table 1.

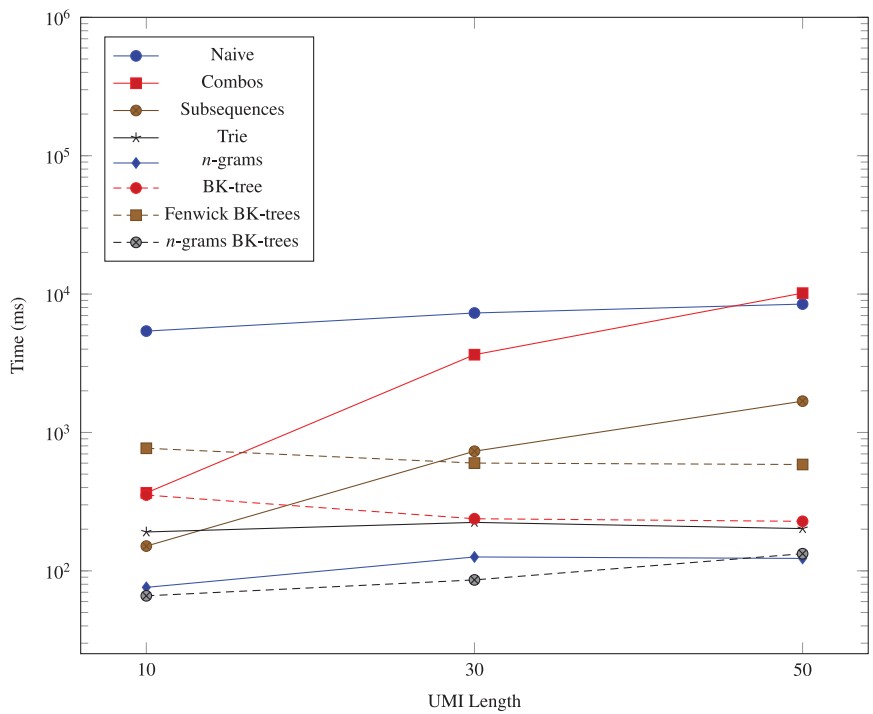

**Figure 5** Time taken (in milliseconds) for different algorithms to run as the UMI length ($M$) changes. The vertical axis is log-scaled.

**Table 2** Statistics of different data structures on a large, simulated dataset with 167,578 unique UMIs.

| Data Structure | Property | Value |
|---|---|---|
| Subsequences | Number of bins | 1,471,978 |
| | Avg bin size | 1.1 |
| | Max bin size | 5 |
| Trie | Nodes | 511,634 |
| $n$-grams | Number of bins | 6,250 |
| | Avg bin size | 53.6 |
| | Max bin size | 139 |
| BK-tree | Avg depth | 8.6 |
| | Max depth | 13 |

We also generate three datasets $(10^3, 10, 1)$, $(10^3, 30, 1)$, and $(10^3, 50, 1)$ for evaluating the performance of each data structure as the UMI length $M$ changes. The result from using these datasets is graphed in Fig. 5.

We generate three more datasets $(10^3, 10, 1)$, $(10^3, 10, 2)$, and $(10^3, 10, 3)$ to measure the performance impact of increasing the number of edits allowed in the data structures. The run times for these datasets are shown in Fig. 6.

We also show some statistics regarding a few selected data structures constructed from the $(10^4, 10, 1)$ dataset in Table 2.

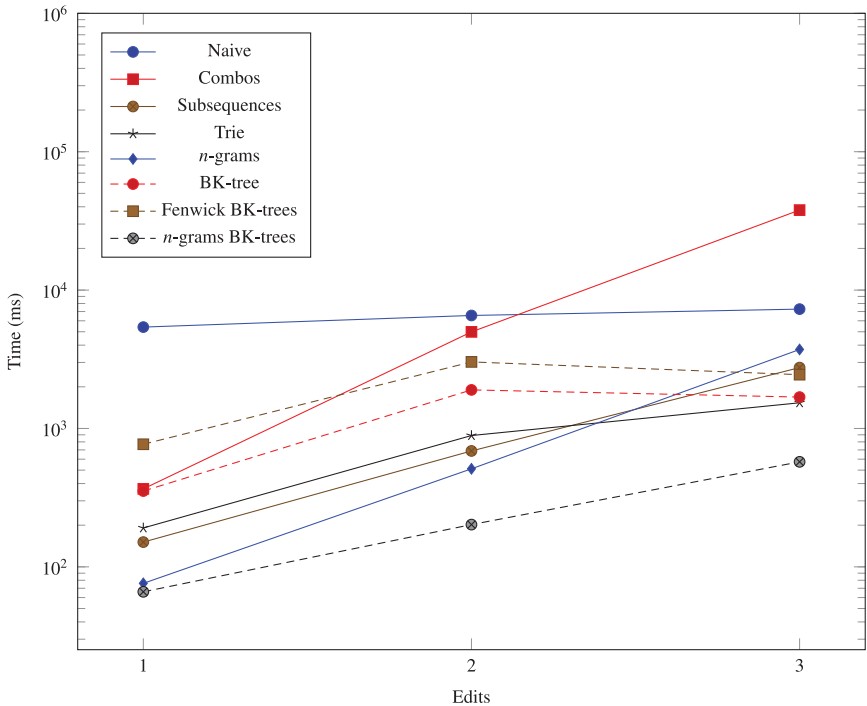

**Figure 6** **Time taken (in milliseconds) for different algorithms to run as the Hamming distance threshold ($k$) changes.** The vertical axis is log-scaled.

**Table 3** **Time taken to deduplicate a BAM file with many UMIs spread out over different alignment coordinates.** The $n$-grams BK-trees data structure is used in UMICollapse. Run time is measured in seconds. Default settings (i.e., directional algorithm, allowing one edit) are used for both UMI-tools and UMICollapse. The best time for each dataset is bolded.

| Dataset | UMIs | UMI-tools | UMICollapse |
|---|---|---|---|
| *Kivioja et al. (2012)* | 1,338,610 | 41.46 | **8.732** |
| *Müller-McNicoll et al. (2016)* | 1,175,027 | 8.69 | **4.26** |
| *Müller-McNicoll et al. (2016)* | 12,925,297 | 94.13 | **34.43** |
| *Müller-McNicoll et al. (2016)* | 118,677,727 | 889.42 | **271.31** |

## Comparison to UMI-tools

We compare our UMICollapse to the dedup function in UMI-tools (*Smith, Heger & Sudbery, 2017*) on two main tasks. The first main task tests the tools on handling large datasets with many UMIs spread out among the alignment positions. One of the datasets we test is the aligned single-end reads from a controlled replicate of *Müller-McNicoll et al. (2016)*, which is also used by UMI-tools (*Smith, Heger & Sudbery, 2017*) as an example dataset. In total, there are 1,175,027 input UMIs of length 9 across all 12,047 alignment positions, and the maximum number of unique UMIs at an alignment location is only 252. The second dataset is the aligned single-end reads from the first replicate of *Kivioja et al. (2012)*, which is used as an example dataset for TRUmiCount (*Pflug & von Haeseler, 2018*). There are 1,338,610 reads across 61,193 alignment coordinates, with

**Table 4 Time taken to deduplicate a BAM file with many UMIs at a single alignment coordinate.** The *n*-grams BK-trees data structure is used in UMICollapse. Default settings (i.e., directional algorithm, allowing one edit) are used for both UMI-tools and UMICollapse. Run time is measured in seconds. UMI-tools was terminated on larger datasets since it took longer than 20 min to run on those datasets. The best time for each dataset is bolded.

| Unique UMIs | UMI-tools | UMICollapse |
|---|---|---|
| 16,392 | 5.48 | **0.48** |
| 158,393 | >1,200 | **3.09** |
| 1,114,686 | >1,200 | **26.36** |

a maximum of 8,227 unique UMIs of length 10 at any single alignment coordinate. The aligned UMI data used in the experiments (example UMI-tools data) is available at https://github.com/CGATOxford/UMI-tools/releases/download/1.0.0/example.bam. The other aligned UMI dataset used in the experiments (example TRUmiCount data) is available at https://cibiv.github.io/trumicount/kv_1000g.bam.

We also create two larger datasets where each UMI in the (*Müller-McNicoll et al., 2016*) example dataset is amplified with one substitution edit 10 and 100 times. The two datasets contain 12,925,297 and 118,677,727 UMIs, respectively, and the run times of both tools on all four datasets are shown in Table 3.

The second main task tests whether UMICollapse represents an improvement over UMI-tools (*Smith, Heger & Sudbery, 2017*) on the data structures used for accelerating the network-based deduplication algorithms. We simulate three datasets with many unique UMIs at the exact same alignment position. UMIs are generated by first generating $C$ random center UMIs of length 9 and then generating 20 random non-center UMIs for each center UMI, where each non-center UMI is within 1 edit of its corresponding center UMI. Our three datasets of $C = 10^3$, $C = 10^4$, and $C = 10^5$ result in 16,392, 158,393, and 1,114,686 unique UMIs at a single alignment position, respectively. The timings of both tools are shown in Table 4.

Since our proposed methods for speeding up the UMI deduplication task do not change the result of the graph-based UMI collapsing algorithms, UMICollapse and UMI-tools (*Smith, Heger & Sudbery, 2017*) should output the exact same results. In practice, the only difference between the two tools' results is the UMI that is chosen to be outputted when there are ties in the UMI frequencies of multiple UMIs. The UMI that is selected is not well-defined for UMI-tools (*Smith, Heger & Sudbery, 2017*) and UMICollapse, so an arbitrary UMI may be chosen. However, this rarely occurs in the two deduplication tasks that we evaluate.

## DISCUSSION

The data structures we examine all run significantly faster than the naive $O(N^2)$ method. The *n*-grams BK-trees data structure performs very well even when the number of unique UMIs, the UMI length, and the number of edits increases. In some cases, it is almost 100 times faster than the naive method. As the number of unique UMIs increase, methods like combo, subsequences, and trie, which rely on generating nearby UMIs within a certain edit

distance, begin to catch up to the *n*-grams BK-trees method. However, those methods do not scale well as the UMI lengths and the number of edits increases. The Fenwick BK-trees data structure does not perform well due to the overhead of maintaining a Fenwick tree across multiple BK-trees, and the *n*-grams BK-trees method represents a direct improvement over the BK-tree data structure and the *n*-grams data structure, individually. Therefore, the data structure that performs well in all situations is the *n*-grams BK-trees method.

In terms of theoretical memory usage, the subsequences method and the trie data structure are among the most memory inefficient. Many subsequences must be generated and stored in the subsequence method (e.g., with 167,578 unique UMIs, 1,471,978 unique subsequences are stored in a hash table), which takes up a significant portion of memory. The trie method requires many nodes to represent the UMI sequences (e.g., with 167,578 unique UMIs, 511,634 trie nodes must be created), which results in a significant memory overhead to store each trie node and the pointers to its children in memory.

From the experiment with over $10^8$ input reads spread out across all alignment coordinates, we find that both UMI-tools (*Smith, Heger & Sudbery, 2017*) and UMICollapse are capable of handling extremely large input datasets. UMICollapse is consistently around 3 times faster than UMI-tools (*Smith, Heger & Sudbery, 2017*) as the number of input UMIs increases across all alignment positions. Since there is only a constant factor speed difference as the number of UMIs increases across all alignment coordinates, this difference can be mostly attributed to the differing programming languages and other implementation details. Therefore, the speed benefit of using a complex data structure like *n*-grams BK-trees is not present in this task. The algorithms we discuss are for increasing the speed of deduplication as the number of unique UMIs increase at *one* alignment location, so this experiment is a poor evaluation task for our tool.

As the number of unique UMIs at one single alignment position increases to over $10^6$, UMI-tools (*Smith, Heger & Sudbery, 2017*) takes more than 20 min, while UMICollapse finishes in only 26 s. The speed difference here is orders of magnitude larger than the 3 times improvement in the previous experiment with UMI sequences distributed across multiple alignment positions. This means that the bottleneck in the deduplication process of UMI-tools (*Smith, Heger & Sudbery, 2017*) is indeed within the edit distance computations across pairs of UMI sequences, and UMICollapse is much more efficient in this aspect. We expect UMICollapse to be much faster than other tools that must compute pairwise ($O(N^2)$) UMI edit distances.

## CONCLUSION

The UMI deduplication problem can be formulated in a manner that enables optimizations to be made. Instead of static pairwise edit distance computations, we formulate the problem as a query problem on a dynamic graph that involves a common interface, and we show that previous deduplication algorithms can be implemented with this interface no change to their results. We also propose multiple data structures that implements this interface, and we find that the *n*-grams BK-trees data structure is the most efficient through an empirical evaluation with simulated datasets. We implement our algorithms in a proof-of-concept

tool called UMICollapse, and we find that it is indeed faster than the popular UMI-tools (*Smith, Heger & Sudbery, 2017*) by a wide margin on large, simulated datasets.

### Funding

The author received no funding for this work.

### Competing Interests

The author declares there are no competing interests.

### Author Contributions

- Daniel Liu conceived and designed the experiments, performed the experiments, analyzed the data, contributed reagents/materials/analysis tools, prepared figures and/or tables, authored or reviewed drafts of the paper, approved the final draft.

### Data Availability

The source code for UMICollapse is available at GitHub: https://github.com/Daniel-Liu-c0deb0t/UMICollapse.

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
