# Peer review of "Algorithms for efficiently collapsing reads with Unique Molecular Identifiers"

_PeerJ, doi:10.7717/peerj.8275_

## Round 0.1 · original submission · Major Revisions

The two reviewers give a lot of constructive and insightful comments, whose consideration will substantially improve the manuscript. In fact, many of these are fairly minor issues that can be resolved by adding a few details to the text. Other requests (some considered optional by reviewer 2, but still relevant) will require more work, which is why I classify this as "major revisions" (instead of minor revisions). It is clear, overall, that this is insightful novel research with immediate use for several applications, and I am confident that we can accept the paper quickly once said issues have been addressed in a revision.

Reviewer 1 ·

Basic reporting

* Lines 11-12, the sentence "reads on their finding reads" is unclear and should be rephrased.
* Line 15, the author says "We formulate the problem of deduplicating UMIs", I'd advise using the word "reformulate" as the problem of UMI deduplication was already formulated by Smith et. al. [1] on which the author is proposing the improvements.
* Lines 17-18, the author says "which can deduplicate over one million unique UMIs of length 9 at a single alignment position in around 26 seconds.", this sounds great however it's unclear how much computational resource like threads / CPU / memory was used.
* Line 23-24, the author says "By ligating a short, random UMI sequence onto each strand of DNA fragment before PCR amplification, sequenced reads with the same UMI can be easily identified as PCR duplicates.". The author has attempted to summarize a lot of details into one sentence making it dense to read. This can use a little more detail as it motivates the usefulness of the problem of UMI deduplication of which the author is proposing the solution.
* Line 26, Author should cite a paper which uses iClip data for UMI deduplication.
* Line 28-29, "However, due to sequencing and PCR amplification errors, sequenced reads may contain UMI sequences that deviate from their true UMI sequences before PCR amplification", Author should cite relevant paper for this claim.
* Line 33-34 "they must first be aligned to a reference genome, ", this claim is not true, Vasilis et al. [2] and Srivastava et al. [3] aligns to transcriptome for performing the deduplication.
* Line 36, 47, 244 multiple latex citation issues " umis [umis], gencore [2], fgbio [fgbio], Picard [Picard Tools]" and at many other places. The author should recheck the manuscript for all such errors.
* Line 40-42, It's unclear if the author is talking about single-cell UMI deduplication or in-general PCR deduplication of reads. Author's claim " a consensus read must be extracted for each group of reads" is not always true for the single-cell for example in [2, 3] author generates the gene count directly i.e. they don't assign a consensus to read. Assigning "consensus read" and "read counting" are two different yet related computational problem.
* Line 44, "many works focused on discussing algorithms that make the UMI deduplication process more efficient.", It's unclear which efficiency the author is talking about accuracy, resource usage?
* Line 48-49 Starting of the sentence should be as Smith et. al. [18] instead of just the citation number. Also, the sentence is incomplete " in their UMI-tools *method*".
* Line 50 "That allows it to avoid the issue of overestimating the number of true UMIs from directly counting unique UMIs.", why is there an issue of overestimation, needs more elaboration?
* Line 51, "still being error-tolerant" what kind of errors, both SNP and indels?
* Line 54, "For simplicity, we only consider the problem of deduplicating UMIs at a single alignment position." what are the implications of these assumptions ? can it be generalized?
* Line 70, "Most deduplication algorithms attempt to build graphs", cite one such algorithm.
* Line 72-74, "There are three main deduplication algorithms for identifying groups of unique UMIs based on G", Author should be more clear about the mentioned graph-based algorithms here as Srivastava et al. [3] also proposed graph-based deduplication algorithm which is similar to "directional" but not same and is not mentioned in this context.
* Line 79, "and v is not removed", its unclear v is not removed from where? I think this sentence can use some rephrasing to make it more clear.
* Line 150, "Most of them are special data structures that speed up REMOVE NEAR", the data structure is "special" in what sense? It seems unnecessary superlative as the data-structures used were previously used in other problems too having said that I don't mean to question the novelty of the application.
* Line 228-234, More clarity can be added by stating that the n-gram approach follows the pigeonhole principle [4].
* Line 416, It's unclear what does other means by dynamic query problem?
* Line 418, "implemented with this interface no change to their results. " It's unclear to me if the author wants to sell the paper as a proof-of-concept or wants users to adopt the data structure library built in Java. If later, then there should ideally be some form of evidence be provided that compares the output of UMI-tools and UmiCollapse suggesting there is no difference in the generated output, it doesn't have to be formally shown in the paper but I'd recommend having the analysis made available may be as an ipython notebook?
* Table 1. It is fine to use `Combo` and `Subseq.` as the shorter representation of Combination and Subsequence but it should be mentioned in the caption or the text the meaning of these shorthands.
* Multiple problems in the References like line 466, 467.


[1] Smith, Tom, Andreas Heger, and Ian Sudbery. "UMI-tools: modeling sequencing errors in Unique Molecular Identifiers to improve quantification accuracy." Genome Research 27.3 (2017): 491-499.
[2] Ntranos, Vasilis, et al. "Fast and accurate single-cell RNA-seq analysis by clustering of transcript-compatibility counts." Genome Biology 17.1 (2016): 112.
[3] Srivastava, Avi, et al. "Alevin efficiently estimates accurate gene abundances from dscRNA-seq data." Genome Biology 20.1 (2019): 65.
[4] https://en.wikipedia.org/wiki/Pigeonhole_principle

Experimental design

* Although not a requirement but the manuscript could benefit from an independent validation of the simulated data generated using the tool from Sarkar et. al. [5].

[5] Sarkar, Hirak, Avi Srivastava, and Rob Patro. "Minnow: a principled framework for rapid simulation of dscRNA-seq data at the read level." Bioinformatics 35.14 (2019): i136-i144.

Validity of the findings

no comment

Additional comments

Author discusses the reformulation of the UMI-deduplication problem using efficient data structure which improves the overall running time of the current algorithm by a significant margin. Apart from minor edits in the background section, I find the paper to be very well written however author should consider revising the manuscript with some more experienced latex user as there are multiple syntactical problems throughout the manuscript. I find the methodological contribution in the manuscript to be very interesting and useful for the community but I am little curious about the validations as they are performed on a custom simulations. The manuscript could benefit from an independent validation of the simulated data generated through already published tools.

·

Basic reporting

The manuscript is excellently written, with good, professional English through-out. The text is to the point, avoiding unnecessary verbosity without excluding the relevant explanation. There is a sufficient review of the literature to allow the reader to appreciate the problem at hand and the work proceeding this one. One small quibble, the author cites Schirmer et al (reference 16) as an example of previous work on the UMI deduplication task, but although the paper does use UMIs in one of its library preparation methods, it doesn’t really examine the task of deduplicating reads based on these. The references are in an odd style, with numbers for peer-review literature references, and names for citations to grey-literature such as github repositories. However, I understand the journal will reformat these? Figures and tables are well presented, with a sufficient font size, and acceptable legends. There is little “raw data” to speak of for this work and so raw data sharing is not really relevant.

Experimental design

The work is primary research, investigating the performance of a range of different algorithms/data structures, some novel, and thus fits within the Aims and Scope of the journal. The research question (which data structures allow the fastest performance in the UMI deduplication problem) is well defined and the study clearly addresses the question.

The methods are well described in terms of the algorithms that the author implements, but parts of the process are less well described. I don’t see any reference to code used in the study beyond the code of the tool itself. For example, the author might want to include the code that generated the simulated datasets used in the study. Benchmarking takes place against UMI-tools, but the author does not describe the version of UMI-tools used, nor the command-line. This is important, because, as the author notes, early version of UMI-tools used the “naïve” approach, while later versions implemented a version of the n-grams structure described in the manuscript. Also various command line setting can have a large effect on the performance of a tool.

Validity of the findings

The author compare their different algorithms to each other and to UMI-tools on simulated data. UMICollapse and UMI-tools are also compared on two real datasets. One thing which I did not understand about the simulated datasets is the number of unique UMIs. For example, the authors state that they generate a set of simulated UMIs by taking 10^5 different 10-bp UMI sequences and generating 20 random sequences that are within 1 edit of each of these 10^5 “centre” UMIs. The author says this generates 1,550,871 unique UMIs. But unless I’m mistaken, there are only 1,048,576 possible 10nt UMI sequences. Is each “centre” UMI and its child sequence fed into the algorithm separately. If so this should be explained. This seems unlikely however as the data in Table 4 is described as “many unique UMIs at the exact sam alignment position”, but has one position with 1,114,686 unique 9nt UMIs, when there are only 262,144 unique 9nt sequences. More likely the author has counted some sequences twice when a single change from two centre UMI results in the same new sequence.

The author might also like to comment on the method for generating the simulated datasets, which results in a particular structure, where each “child” UMI is only one hop away from the parent UMI, and there are no chain of child UMIs, such as AAAA -> AAAT -> AATT, or other, more complex network structures. Might this be why UMI-tools is only 3x slower than UMICollapse in the data presented in Table 3, where constructing the network might account for less of the time spent?

The author convincingly shows that on the simulated dataset, at a single position, the n-grams and BK-tree structures outperform other structures, and that their combination, n-gram BK-tree performs best of all, being 2x as fast as n-grams and 20x as faster as BK-tree on the largest number of UMIs tested on all three. UMICollapse is also orders of magnitude faster than UMI-tools on simulated data from a single, highly dense, alignment location. If this is due to the algorithm/data structure used, then it is surprising because UMI-tools uses a version n-grams, which Table 1 shows to only take 2x as long as the n-grams BK structure used for UMICollapse in Table 4. Thus I don’t think the claim that the massive improvement over UMI-tools is due to the data structure used can be entirely supported. What role might other factors, such as the edit_distance calculation routine, the efficiency of code implementing the same algorithms or the choice of programming language make in this difference?

The authors do not report whether their algorithm returns the same results as UMI-tools. Although not returning the exact same reads is understandable, the author should report if the same or similar number of reads is returned for both the simulated and real datasets.

The discussion mentions the memory efficiency of the different algorithms, but no data is presented in the manuscript. The author should make clear if the discussion of memory usage is based on a theoretical assessment of how much memory the different algorithms should use, or a measurement of the actual memory usage.

Additional comments

In several use cases it is useful to not just select and output a single read for each group of reads with a similar UMI, but the output all reads, tagged with the “parent UMI”, or UMI group that have been associated with. Because the methods described in this paper imply the UMI network, rather than explicitly constructing it, it is not clear to me if this is possible, particularly if a child UMI is several steps removed from its parent. Could the author clarify which data structures/algorithms would be compatible with this?

In their description of the directional adjacency algorithm, the author refer to the inequality 2f(v)+1≤f(u). In fact the inequality in the UMI-tools paper is 2f(v)-1≤f(u). It is not clear if this is a typo in the paper or if this is in the code.

I have successfully downloaded, installed and run the UMICollapse software using the example data. The authors are to be congratulated on the easy installation.

Summary

The size of datasets currently being used is getting to the point where it is becoming impossible to use error correcting UMI deduplication algorithms due to their time and memory requirements. This is thus an important and timely contribution. The author is to be congratulated on a through and insightful study. I recommend publication with minor alterations. In order of importance:
1. The author should clarify the status of the frequency inequality.
2. The author should explore the similarity/compatibility of the results from UMICollapse and UMI-tools as well as just the timings.
3. The author should clarify the number of unique UMIs in the simulated datasets
4. The author should re-consider their claim that the bottleneck in UMI-tools is definitely in the edit distance calculations, and at least discuss the fact that their own computations show that the n-grams bk-tree structure is only 2x fast than the n-gram structure, but UMICollapse appears to be orders of magnitude faster than UMI-tools.
5. The methods, outside the description of the data structures/algorithms should be more detailed. Most importantly the version of UMI-tools must be specified.

The following would improve the manuscript but might be considered optional:
1. The author could discuss the scheme for simulating data and how that might affect their comparisons.
2. The author could discuss if these algorithms are suitable for assigning a parent UMI to every read and outputting that as a tag on the read.
3. The authors should clarify the status of the memory usage estimates, and include real usage estimates for the difference algorithms if possible

---

## Round 0.2 · Minor Revisions

In principle, both reviewers agree that the paper can be accepted as it is, but reviewer 2 requests more explanations about the handling of unspecified nucleotides (Ns). If you could add details about this and write a short summary of these changes, no full round of review will be necessary after the next revision and the revised work can be accepted quickly.

Reviewer 1 ·

Basic reporting

All my major concerns have been addressed by the author.

Experimental design

N/A

Validity of the findings

N/A

Additional comments

I'd like to thank and congratulate the author for thoroughly replying and improving the manuscript. I think the manuscript is in a better shape and ready to publish, provided the formatting is fixed by the journal.

·

Basic reporting

No comment

Experimental design

The author has clarified that the reason for the large number of UMIs at a single positions is that they simulate sequences from {A,T,G,C,N}. Could they common on how N bases are handled in edit distance calculations? Is N to any base seen as a match or a mismatch? They say that there bit-shifting edit distance method uses 3-bits per base, but this must be different for UMIs that contain Ns? Indeed, none of the discussion of methods seems to recognize that Ns might be included.

Validity of the findings

No comment

Additional comments

The author has addressed most or all of my previous comments. Apart from the minor quibble above I believe the manuscript is now ready to publish. If the above is clarified I do not envisage it being necessary to review the manuscript again.

---

## Round 0.3 · accepted · Accept

Thank you for addressing the unspecified N nucleotides in your review. I am assuming your comment means that N is different from ACGT, but N equals other Ns. An alternative would be that N equals nothing, not even itself.